# Pin1 as Molecular Switch in Vascular Endothelium: Notes on Its Putative Role in Age-Associated Vascular Diseases

**DOI:** 10.3390/cells10123287

**Published:** 2021-11-24

**Authors:** Francesca Fagiani, Marieva Vlachou, Daniele Di Marino, Ilaria Canobbio, Alice Romagnoli, Marco Racchi, Stefano Govoni, Cristina Lanni

**Affiliations:** 1Pharmacology Section, Department of Drug Sciences, University of Pavia, V.le Taramelli 14, 27100 Pavia, Italy; francesca.fagiani@unipv.it (F.F.); marieva1912@hotmail.com (M.V.); racchi@unipv.it (M.R.); cristina.lanni@unipv.it (C.L.); 2Department of Life and Environmental Sciences, Polytechnic University of Marche, Via Brecce Bianche, 60131 Ancona, Italy; d.dimarino@staff.univpm.it (D.D.M.); a.romagnoli@staff.univpm.it (A.R.); 3New York-Marche Structural Biology Center (NY-MaSBiC), Polytechnic University of Marche, Via Brecce Bianche, 60131 Ancona, Italy; 4Department of Biology and Biotechnology, University of Pavia, V.le Taramelli 14, 27100 Pavia, Italy; ilaria.canobbio@unipv.it

**Keywords:** Pin1, vascular endothelium, nitric oxide synthase, nitric oxide, vascular dementia, aging

## Abstract

By controlling the change of the backbones of several cellular substrates, the peptidyl-prolyl *cis-trans* isomerase Pin1 acts as key fine-tuner and amplifier of multiple signaling pathways, thereby inducing several biological consequences, both in physiological and pathological conditions. Data from the literature indicate a prominent role of Pin1 in the regulating of vascular homeostasis. In this review, we will critically dissect Pin1’s role as conformational switch regulating the homeostasis of vascular endothelium, by specifically modulating nitric oxide (NO) bioavailability. In this regard, Pin1 has been reported to directly control NO production by interacting with bovine endothelial nitric oxide synthase (eNOS) at Ser^116^-Pro^117^ (human equivalent is Ser^114^-Pro^115^) in a phosphorylation-dependent manner, regulating its catalytic activity, as well as by regulating other intracellular players, such as VEGF and TGF-β, thereby impinging upon NO release. Furthermore, since Pin1 has been found to act as a critical driver of vascular cell proliferation, apoptosis, and inflammation, with implication in many vascular diseases (e.g., diabetes, atherosclerosis, hypertension, and cardiac hypertrophy), evidence indicating that Pin1 may serve a pivotal role in vascular endothelium will be discussed. Understanding the role of Pin1 in vascular homeostasis is crucial in terms of finding a new possible therapeutic player and target in vascular pathologies, including those affecting the elderly (such as small and large vessel diseases and vascular dementia) or those promoting the full expression of neurodegenerative dementing diseases.

## 1. Aging and the Vasculature: Focus on Endothelial Dysfunction

The vascular endothelium, the active inner layer of the blood vessel, releases a wide array of biologically active molecules acting in an autocrine or paracrine fashion, thereby controlling arterial structure and vasodilatory, thrombolytic, and vaso-protective functions. In particular, it regulates a number of biological functions, such as substrate exchange/transport, innate immunity, the regulation of vascular tone by balancing the production of vasodilators and vasoconstrictors, angiogenesis, and hemostasis by secreting antiplatelet and anticoagulant molecules (for a comprehensive review on the topic see [1]). Endothelial dysfunction occurs during aging [2], where an imbalance between vasodilator and vasoconstriction factors, a progressive reduction of nitric oxide (NO) bioavailability, as well as an increase in cyclooxygenase (COX)-derived vasoconstrictor factors have been reported [2,3]. In addition, a decreased expression and activity of endothelial NOS (eNOS) has been observed in older animals [4]. Moreover, endothelial dysfunction represents a risk factor for the development of several human vascular diseases, such as atherosclerosis, hypertension and stroke, peripheral arterial disease, metabolic syndrome (obesity, insulin resistance), and diabetes. The molecular mechanisms underpinning endothelial dysfunction are rather complex. Indeed, multiple mechanisms (i.e., impaired vasodilation, oxidative stress, inflammation, cell injury/death, senescence) have been reported to be involved. Notably, among them, decreased NO bioavailability represents a key hallmark of endothelial dysfunction. Thus, the severity of endothelial dysfunction has been shown to have prognostic value for cardiovascular events [5].

## 2. The Role of the *cis-trans* Isomerase Pin1 in Vascular Endothelium

Originally identified as a protein physically interacting with NIMA (never in mitosis A) mitotic kinase, Pin1 (protein interacting with NIMA-1) is a member of the evolutionarily conserved peptidyl-prolyl *cis-trans* isomerase (PPIase) family that, unlike all other known prolyl isomerases, mediates the isomerization of phosphorylated serine- and threonine-proline (pSer/Thr-Pro) motifs, mediating the conformational change of cellular substrates [6]. From a structural point of view, human Pin1 contains an N-terminal WW protein interaction domain (residues 1–39) and a C-terminal PPIase domain (residues 50–163), connected by a flexible linker [6] (Figure 1A). Notably, Pin1-catalyzed phosphorylation-dependent conformational changes of its substrates have been reported to control many key regulators involved in numerous cellular processes (e.g., cell growth, cell cycle progression, cellular stress response, development, apoptosis, neuronal differentiation, and immune response) [7]. In particular, it acts as a unique molecular timer regulating multiple targets at different steps of a given signaling pathway to synergistically orchestrate cellular responses [7]. 

Notably, Pin1 deregulation has been associated with a number of pathological conditions. In particular, it has been demonstrated to promote oncogenesis by modulating several oncogenic signaling pathways and its overexpression has been shown to correlate with poor clinical outcome [8]. Moreover, Pin1 deregulation has been observed in age-related and neurodegenerative diseases, including Alzheimer’s disease (AD), Parkinson disease, frontotemporal dementia, Huntington disease, and amyotrophic lateral sclerosis, where it mediates profoundly different effects, ranging from neuroprotective to neurotoxic (for a review on the topic see [9]).

Pin1 has been found to act as a critical driver of vascular cell proliferation, apoptosis, and inflammation, with implication in many cardiovascular diseases (e.g., atherosclerosis, coronary restenosis, and cardiac hypertrophy) [10,11,12]. Notably, Pin1 has been reported to regulate the degradation of the inducible nitric oxide synthase (iNOS) in murine aortic endothelial cells upon induction by lipopolysaccharide and IFN-γ [13], as well as to interact with eNOS in a phosphorylation-dependent manner, thereby suppressing eNOS activity, similar to the tonic suppression of eNOS activity by caveolin-1 [14]. However, despite sparse evidence showing the implication of Pin1 in vascular homeostasis, little is known about Pin1 implication in cerebrovascular homeostasis, both in physiological and pathological contexts. As an example, Pin1 distribution and intracellular compartmentalization in the cerebrovascular system remains unknown. In the following sections, we will discuss evidence from the literature indicating the role of Pin1 in the regulation of vasculature homeostasis, by specifically dissecting its molecular interactions with key intracellular players, such as NOS, as well as the implication of Pin1 in vascular pathology. Moreover, we will discuss the hypothesis that Pin1 may serve a pivotal role in the function of vasculature, mainly focusing on its effects on the vascular endothelium. This latter, which is at direct contact with blood and affected by various vascular risk factors, represents a critical target and key player in vascular diseases. 

Therefore, understanding the role of Pin1 in the vascular homeostasis and specifically, its interplay with NOS activity, is crucial in terms of finding a new possible therapeutic player and/or target in vascular pathologies, such as hypertension, diabetes, small and large vessel diseases, and vascular dementia.

## 3. Multi-Way Regulation of NO Production by Pin1

### 3.1. Direct Regulation of eNOS Activity by Pin1

Data from the literature indicate that neuronal communication, as well as blood vessel modulation and immune response, are regulated by NO, generated from arginine by a family of three distinct calmodulin-dependent NOS enzymes [15]. In humans, three different isozymes of NOS are encoded by genes located on chromosomes 12, 17, and 7: neuronal NOS (nNOS or NOSI), endothelial NOS (eNOS or NOSIII), both constitutively expressed enzymes stimulated by increases in intracellular calcium, and an inducible NOS (iNOS or NOSII). In particular, eNOS is mainly involved in the regulation of vascular and cerebrovascular endothelial cells, where it releases the vasodilating and vaso-protective NO, regulating the function of blood vessels and contributing to the basal tone of arteries and the control of blood pressure [16]. Notably, eNOS is post-transcriptionally regulated by reversible phosphorylation and protein–protein interactions [17]. The multi-site phosphorylation of eNOS occurs in response to a wide plethora of stimuli and involves several kinases and phosphatases. Phosphorylation of human eNOS at Ser^1177^ (bovine equivalent Ser^1179^) and Ser^633^ (bovine equivalent Ser^635^) is fundamental for eNOS activation, while phosphorylation at Thr^495^ (bovine equivalent Thr^497^) in the Ca^2+^/calmodulin binding domain is inhibitory [17]. The effects of phosphorylation at eNOS-Ser^114^ (bovine equivalent Ser^116^) remain controversial. Notably, Pin1 has been reported to interact with eNOS in a phosphorylation-dependent manner [14,18]. Accordingly, Ruan et al. demonstrated a Ser^116^ phosphorylation-dependent interaction of bovine eNOS with Pin1 in bovine endothelial cells by immunoprecipitation [14]. The blockage of Ser^116^ phosphorylation reduced the interaction between eNOS and Pin1, indicating that such physical binding occurs in a site-specific and phosphorylation-dependent manner. The phosphorylation-dependent binding of Pin1 on eNOS was further confirmed in COS-7 cells, a fibroblast-like cell line not endogenously expressing eNOS, that were transfected with either wild-type bovine eNOS (WT eNOS) or pSer^116^-mimetic form of bovine eNOS (S116D eNOS) [14]. In this latter, aspartic acid, due to its chemical similarity to pSer, replaced Ser^116^ by site-directed mutagenesis. Full-length human Pin1 cDNA was inserted in the eNOS transfected cells and the results showed that Pin1 had an increased binding affinity for the S116D eNOS compared to WT eNOS, underscoring the importance of Ser^116^ phosphorylation on Pin1-eNOS interaction [14]. These results indicating that Pin1 binding site on bovine eNOS is at Ser^116^-Pro^117^, whose human equivalent is Ser^114^-Pro^115^, were further confirmed in rat aortic endothelial cells [18]. 

From a structural point of view, human eNOS is composed by two well separated domains, an N-terminal heme-containing oxygenase domain and a C-terminal reductase domain (Figure 1B). The oxygenase domain is the best studied and characterized one from a structural point of view; a total of nine different structural studies that describe several structures are available in the protein data bank (https://www.rcsb.org/; accessed on 27 October 2021), covering about 410–430 amino acids representing almost the entire oxygenase domain. Residues Ser^114^-Pro^115^ belonging to this domain are involved in the interaction with Pin1 and are located on a pretty long solvent-exposed loop (Figure 1B). Part of this loop, comprising residues Ser^114^-Pro^115^, is missing in the majority of the structures available, indicating that the high degree of flexibility of the loop may be a key element able to facilitate the interaction with different molecular partners and among them, Pin1.

**Figure 1 cells-10-03287-f001:**
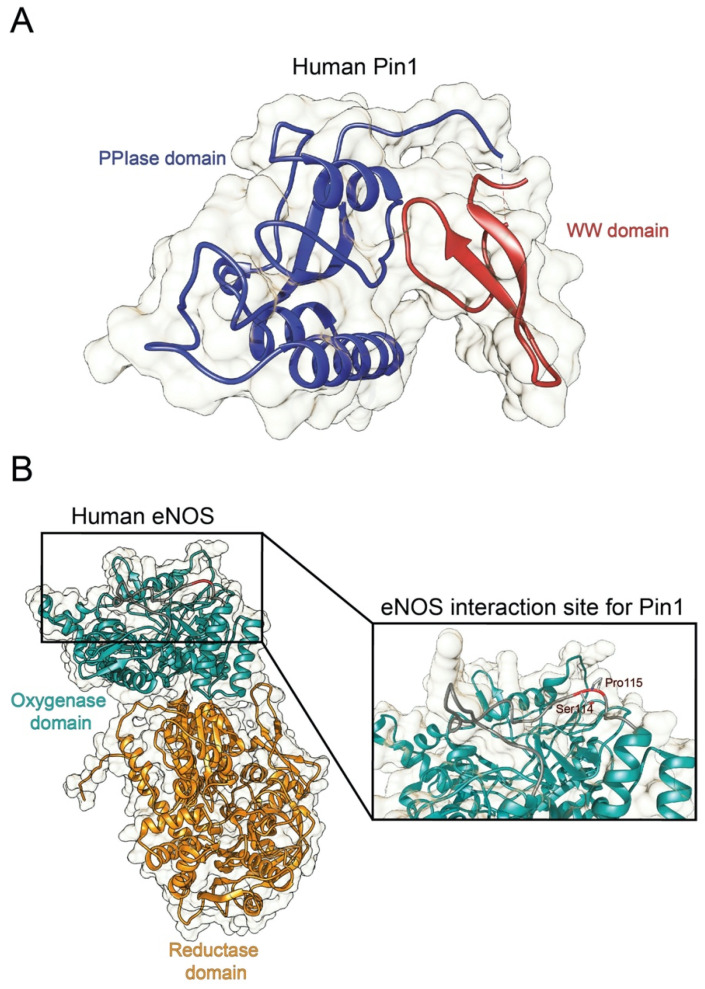
Structure of human Pin1 and human eNOS. (**A**) Ribbon representation of the full structure of human Pin1 (PDBID 1PIN; DOI: 10.1016/s0092-8674(00)80273-1). The PPIase and the WW domain are colored in blue and red, respectively. (**B**) Ribbon representation of the full structure of human eNOS. The structure was recovered from Alphafold protein structure database [19]. The oxygenase and the reductase domains are colored in light green and in orange, respectively. The eNOS Ser^114^-Pro^115^ residues are reported in the inset.

Noteworthy, such Pin1-catalyzed phosphorylation-dependent structural changes on eNOS have been reported to impact eNOS biological activity. In particular, Pin1 binding to eNOS at pSer^116^ has been shown to inhibit eNOS activity. Accordingly, inhibition of Pin1 either with juglone (i.e., a Pin1-specific pharmacological inhibitor) or with a dominant negative Pin1 mutant, enhanced both basal and agonist-stimulated NO production in endothelial cells [14]. Instead, overexpression of wild-type Pin1 in bovine aortic endothelial cells (BAECs) reduced NO production [14]. Moreover, in intact mice aorta, Pin1 overexpression impaired relaxant responses to acetylcholine of aortic rings, compared with rings from control mice [14]. The reduction of the enzymatic activity of purified bovine S116D eNOS was further observed after incubation with purified full-length human Pin1. In line with such evidence reporting a negative regulation of eNOS activity by Pin1, Kennard et al. demonstrated that TNFα treatment of BAECs promoted complex formation between eNOS and Pin1 and suppressed NO secretion [20]. This reduction was completely antagonized by pharmacological inhibition of Pin1 and with transduction with an adenovirus expressing the dominant negative form of Pin1 (Ad-DN Pin1) [20].

In contrast with these findings, Chiasson et al. support the notion that Pin1 positively affects eNOS activity by enabling Ser^116^ dephosphorylation and subsequently enhancing NO production [18]. Consistently, siRNA-mediated knockdown of Pin1 has been found to increase eNOS phosphorylation at Ser^116^, to reduce eNOS phosphorylation at Ser^1177^, required for eNOS activity, and to enhance eNOS Thr^495^ phosphorylation, fundamental for the inhibition of eNOS activity, negatively affecting NO production [18]. Therefore, the increase in phosphorylation of Ser^116^ and Thr^495^, inhibitory sites, and the decrease in phosphorylation of Ser^1177^, stimulatory site, induced by Pin1 deficiency, indicates that the primary mechanism by which NO production decreases relies on the reduction of eNOS activity. Furthermore, Pin1 knock-out mice displayed an enhanced aortic phosphorylation of eNOS at Ser^116^, a reduction in NO production and in endothelium-dependent relaxation responses [18], thus suggesting that Pin1 deficiency-induced endothelial dysfunction and associated hypertension. 

Another study reporting a positive modulation of NOS activity by Pin1, comes from the Liu et al. study reporting that Pin1 interacts with iNOS in murine aortic endothelial cells [13]. Either short hairpin RNA or siRNA knockdown of Pin1 induced an increase in iNOS protein levels and NO production upon treatment with lipopolysaccharide and interferon-γ. However, in the absence of lipopolysaccharide and interferon-γ, Pin1 knockdown did not increase NO production in endothelial cells [13]. 

Taken together, evidence from the literature discussed above is controversial and in particular, two opposite series of results reporting a positive and negative effect by Pin1 on eNOS activity and consequently, on NO production, currently exist (as schematized in Figure 2). In this regard, since Pin1 has been reported to act in a strongly context-dependent manner, such conflicting data may be partially due to the different cellular and tissue environments used in the experimental settings. Indeed, Pin1 functional activity is known to be highly pleiotropic and context-dependent, since it is strictly related to the phosphorylation patterns of its cellular substrates. Indeed, animals and cells may vary both in the basic metabolism of NO and in the regulatory mechanisms, potentially resulting in diverse outcomes in response to different Pin1 levels. Therefore, further investigations are warranted not only to clarify the positive vs. negative role of Pin1 in the regulation of eNOS activity and NO production, but also to understand how alterations of Pin1-mediated effects on NO production may impact vascular homeostasis, thereby triggering vascular pathologies. Moreover, this knowledge is of key importance in order to understand the functional meaning and role of age- and disease-associated changes of Pin1 levels and enzymatic activity described in the literature, specifically when vascular elements are directly involved and/or participate to the expression of the full-blown pathology, as for vascular dementia or Alzheimer’s Disease. 

### 3.2. Indirect Regulation of NOS by Pin1

Beside its role as direct regulator of NOS activity, Pin1 has been also reported to indirectly regulate NO production by interacting with other key intracellular players, critically involved in vascular homeostasis. Accordingly, Pin1 has been suggested to upregulate eNOS activity by modulating the vascular endothelial growth factor (VEGF). VEGF has been largely characterized as a key agonist of eNOS in the vessel wall and as a regulator of eNOS function by triggering the activation of calcineurin, a Ca^+2^- and calmodulin-dependent protein phosphatase (PP2B), which, in turn, dephosphorylates eNOS at Ser^116^ [21]. Kim et al. demonstrated that Pin1 overexpression increased the transcriptional activity and protein levels of VEGF in human MCF-7 cells, by activating the transcription factor of activator protein-1 (AP-1) and the hypoxia-inducible factor-α (HIF-1α) [22]. Thus, Erol et al. speculated that Pin1 may promote VEGF expression that, in turn, triggers the dephosphorylation and activation of eNOS at Ser^116^ [23]. As a proof of concept, a study conducted on humans with bevacizumab (i.e., an anti-VEGF monoclonal antibody) showed that inhibition of VEGF inhibited NO production, promoting endothelial dysfunction and hypertension [24].

Moreover, Pin1 has been demonstrated to inhibit the transforming growth factor-β (TGF-β) signaling [25]. Indeed, Pin1 inhibited TGF-β-driven transcription and gene expression by downregulating Smad2/3 protein levels in COS7 and MDA-MB-231 cells [25]. In particular, Pin1 interacts with Smad2 and Smad3 to increase their association with Smurf2 (Smad ubiquitination regulatory factor 2), leading to Smad ubiquitination and reduced Smad2/3 levels. Notably, activation of TGF-β pathway has been shown to downregulate the expression of eNOS and, consequently, to reduce NO production [26,27]. Thus, it can be speculated that, by inhibiting TGF-β signaling, Pin1 may promote eNOS expression and NO production. However, further studies are needed to confirm such hypothesis. 

## 4. Pin1 Implication in Vascular Diseases

Based on evidence indicating a key role of Pin1 in regulating NO production and vascular homeostasis, alteration of Pin1 levels and/or isomerase activity may be involved in the pathogenesis of vascular pathologies, such as hypertension and diabetes. In this regard, several studies described a prominent role of Pin1 in vascular changes associated to diabetes, where endothelial dysfunction represents an initial process in its vascular manifestations [28]. Paneni et al. demonstrated that treatment of human aortic endothelial cells (HAECs) with high glucose concentration for 72 hours induced an increase in Pin1 mRNA and protein levels [12]. In particular, methylation of Pin1 promoter, which typically represses gene transcription, was substantially decreased during hyperglycemia, suggesting that this epigenetic modification may contribute to Pin1 upregulation [12]. Moreover, during hyperglycemia, Pin1 has been shown to interact with eNOS at Ser^116^ phosphorylation site, thereby promoting eNOS interaction with caveolin-1, an important repressor of eNOS catalytic activity in the endothelium and subsequently reducing NO availability [12]. Interestingly, diabetic Pin1^−/−^ mice were protected against mitochondrial oxidative stress, endothelial dysfunction, and vascular inflammation. In accordance with such evidence, pharmacological inhibition of Pin1 by juglone attenuated endothelial dysfunction, oxidative stress, as well as inflammatory processes, in streptozotocin-induced diabetic mice compared to vehicle-treated animals [29]. Noteworthy, although these data suggest that Pin1 may hinder NO production by repressing eNOS activity during hyperglycemia, it is unknown whether such negative regulation may be ascribed either to a direct effect of Pin1 on eNOS activity or to the increased association of eNOS with caveolin-1 or to a combination of these two effects. 

Moreover, high glucose levels not only triggered Pin1 expression, but also enhanced its enzymatic activity. Consistently, an observational study conducted in patients with type 2 diabetes mellitus showed that Pin1 expression levels and activity were increased in the peripheral blood monocytes of the subjects compared to those of age-matched healthy controls and correlated with glycemic markers (i.e., elevated fasting plasma glucose (FPG) and HbA1C (hemoglobin A1C) levels) [30]. Experiments performed in type 2 diabetic (T2D) mice and in vascular smooth muscle cells exposed to T2D-ressembling conditions also showed that Pin1 levels were increased in hyperglycemic conditions [30]. Finally, Zhang et al. confirmed the notion that high glucose levels up-regulate Pin1, by reporting a substantial elevation in circulatory Pin1 levels in diabetic mice compared to wild-type mice. Such increase was reversed with exposure to juglone [31]. 

## 5. Conclusions

Pin1 has been demonstrated to be a critical regulator in different cellular contexts acting as a molecular timer in the coordination of multiple targets at different steps [2]. An alteration in Pin1 expression and activity has been associated with a number of pathological conditions, ranging from oncogenic signaling pathways, where its overexpression has been shown to correlate with poor clinical outcome [8], to age-related and neurodegenerative diseases, exerting effects ranging from neuroprotective to neurotoxic [8]. Focusing on the vascular homeostasis, Pin1 may drive different outcomes, from vascular cell proliferation, to apoptosis, immune response and inflammation, with implication in many cardiovascular diseases. In this regard, Pin1 also serves a crucial role in controlling vascular inflammation by interacting with specific intracellular targets in circulating cells, as summarized in Table 1. As an example, in peripheral blood mononuclear cells (PBMCs) and activated lymphocytes, Pin1 has been reported to modulate immune response by regulating the mRNA expression and secretion of proinflammatory cytokines, such as granulocyte-macrophage colony-stimulating factor (GM-CSF), IFN-γ and IL-2, and chemokines [32,33,34].

However, the molecular mechanisms through which Pin1 may derange vascular homeostasis remain largely unknown. In this regard, we applied a model-based network approach based on data derived from humans recapitulating the mechanistic connection between Pin1 and the biological pathways involved in vascular processes (Figure 3). A detailed analysis of the network reported in Figure 3 shows a protein–protein interaction (PPI) network between Pin1 and several proteins belonging to other key intracellular pathways. Based on this network, Pin1 interactions with the main players controlling vascular maintenance seems to be only mediated by its action on the transcription factor p53, thus emerging as a not direct interactor. However, no data concerning Pin1 interaction and regulation with the main players in human-derived samples are available. Indeed, the notion about striking differences in animal species and vascular districts highlights the need for studies on patients’ derived samples to confirm the validity of the animal observations in human subjects.

## Figures and Tables

**Figure 2 cells-10-03287-f002:**
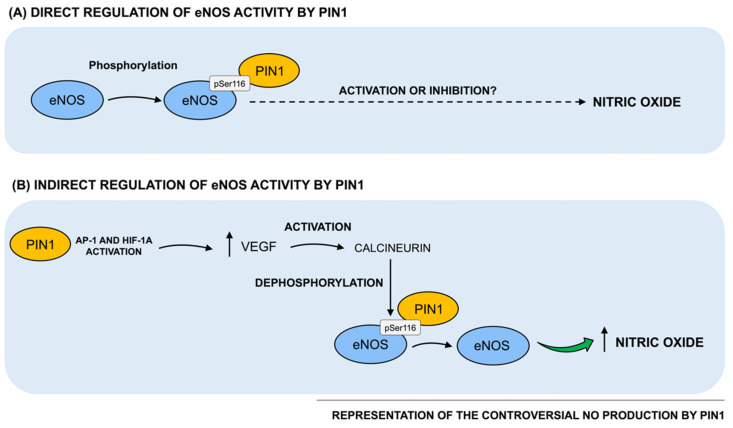
Regulation of Pin1 direct and indirect effects on NO production. (**A**) Notably, Pin1 has been reported to interact with bovine eNOS in a phosphorylation-dependent manner. In particular, Pin1 binding site on bovine eNOS is at Ser^116^-Pro^117^, whose human equivalent is Ser^114^-Pro^115^. Such Pin1-catalyzed phosphorylation-dependent structural changes on eNOS have been reported to consequently impact on eNOS biological activity. However, based on data from the literature, two opposite series of results describing both a positive and a negative regulation of eNOS activity and, consequently, on NO production by Pin1, currently exist. (**B**) Pin1 has been found to indirectly regulate NO production by interacting with other key intracellular players, such as VEGF. Specifically, Pin1 overexpression has been shown to upregulate VEGF transcriptional activity and protein levels, by activating the transcription factor of AP-1 and HIF-1α. It has been hypothesized that Pin1 might increase VEGF expression that, in turn, stimulates the dephosphorylation and activation of eNOS at Ser^116^.

**Figure 3 cells-10-03287-f003:**
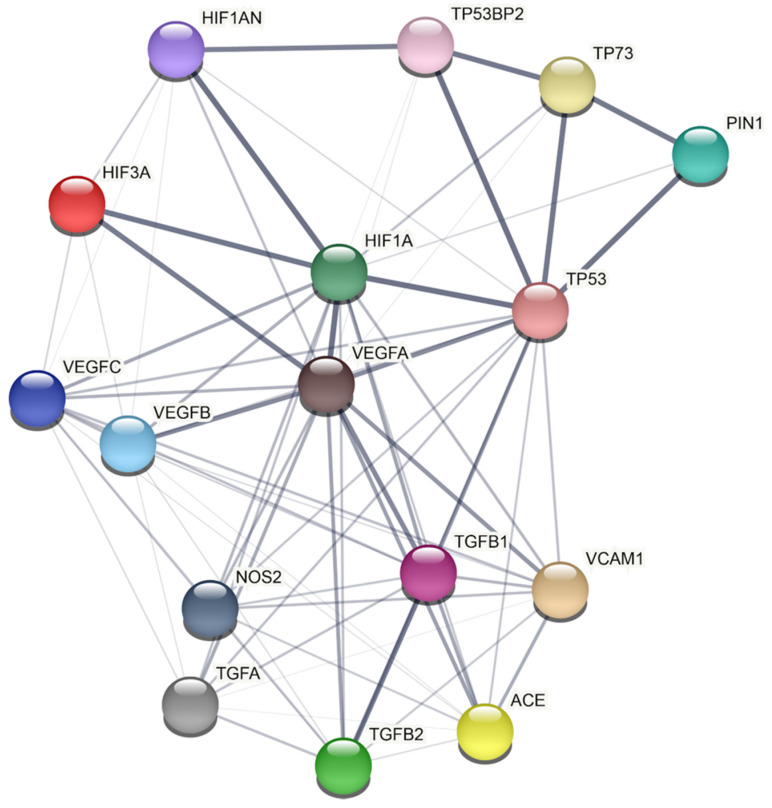
PIN1 protein–protein interaction network by STRING analysis. STRING Protein−Protein Interaction database (http://string-db.org/, accessed on 27 October 2021) (Ver 11.5) has been used to build the PPI network (*Homo sapiens* as chosen organism). Each query protein is individually colored and represented as a node with edged interactions. Network analysis was set at low confidence (STRING score: 0.150). Grey line thickness is an indicator for the strength of data supporting a protein–protein interaction (edges with higher confidence are scored as thicker lines).

**Table 1 cells-10-03287-t001:** Pin1 molecular targets in circulating cells: expanding its role to vascular inflammation.

Cell Types	Molecular Targets	Roles	References
**T Lymphocytes**	A+U-rich RNA-binding factor (**AUF1**)	Pin1 mediates the association of AUF1 with GM-CSF mRNA, which determines the rate of decay by the exosome.	[34]
**T cells**	The transcription factor **PU.1**	Differentiation	[35]
**Eosinophils**	The transcription factor X box-binding protein 1 (**XBP1**) and Interleukin-1 receptor (IL1R)-associated kinase 4 (**IRAK4**) **AUF1** Heterogeneous nuclear ribonucleoprotein C (**hnRNP C**)	Toll-like receptor 7 (**TLR7**)-induced IFN expression. Pin1 mediates association of AUF1 with granulocyte-macrophage colony-stimulating factor (GM-CSF) mRNA, accelerating the rate of decay. Pin1 mediates association of hnRNP C with GM-CSF mRNA, decelerating the rate of decay.	[32,36]
**Neutrophils**	**p47phox** (phox: phagocyte oxidase), the phagocyte NADPH oxidase/NOX2 organizer	Pin1 binds to p47phox, inducing conformational changes that facilitate p47phox phosphorylation by protein kinase C (**PKC**), and results in NADPH oxidase hyperactivation. Pin1 mediates TNF-α–induced neutrophil NADPH oxidase priming and reactive oxigen species hyperproduction via specific binding to p47phox.	[37,38]
**Monocytes**	The transcription factors **RUNX1** and **PU.1**	Pin1 enhances Runx1 activity and represses PU.1 transcription.	[39]
**Macrophages**	**p38MAPK** (p38 mitogen activated protein kinase)	In LPS-induced septic shock, Pin1 indirectly regulates p38MAPK-mediated NLRP3 (NLR Family Pyrin Domain Containing 3) inflammosome.	[40]
**Megakaryocytes**	p-tau	The interaction between Pin1 and p-tau promotes microtubule assembly and proplatelet formation.	[41]
**Endothelial cells**	Tissue factor (**TF**)	The interaction between Pin1 and TF results in increased protein half-life and pro-coagulant activity.	[42]
	NF-Kβ	Deposition of atherosclerotic plaques.	[10]
	eNOS	Pin1 physically interacts with eNOS and inhibits eNOS activation and NO production in BAECs.	[9,14]
	eNOS	Pin1 binds eNOS, promotes eNOS Ser116 dephosphorylation, and increases NO production.	[13]
	iNOS	Pin1 interacts with iNOS and regulates NO production.	[8]
**Vascular endothelial cells**	p53	Pin1 promotes heat stress-induced localization of p53 to mitochondria.	[43]
**Vascular smooth muscle cell (VSMC)**	p53, p21, Gadd45a, p-pRb, p65, and cyclins	In atherosclerotic VSMC Pin1 modulates cellular senescence.	[44]
**Vascular smooth muscle cell (VSMC)**	The transcription factor Bromine domain protein 4, **BRD4**	Pin1 binds BRD4 and regulates proliferation and migration of VSMC.	[45]

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
