# Peer review of "Pin1 as Molecular Switch in Vascular Endothelium: Notes on Its Putative Role in Age-Associated Vascular Diseases"

_cells, 2021, doi:10.3390/cells10123287_

Round 1

Reviewer 1 Report

Fagiani et al. sought to describe molecular charakteristics of Pin-l in vascular biology. Pin-1 has been described an importand mediator for cell proliferation, apoptosis and in inflammation and is
associated with many vascular diseases (e.g. diabetes, atherosclerosis, hypertension, and cardiac hypertrophy).

The manuscript is well written and the description of Pin-1 actions is considerable comprehensive. Though I would think that the manuscript could provide some more details in the context of clinical imprications
in different disease entities (such as diabetes, vascular calcification, sepsis/inflammation, etc.). Also a graphical abstract would be helpful to better put those details in the focus and sum them up for the reader.

Author Response

Point 1: The manuscript is well written and the description of Pin-1 actions is considerable comprehensive. Though I would think that the manuscript could provide some more details in the context of clinical implicationsin different disease entities (such as diabetes, vascular calcification, sepsis/inflammation, etc.). Also a graphical abstract would be helpful to better put those details in the focus and sum them up for the reader.

ANSWER: We thank the Reviewer for the overall positive comments. We rewritten the introduction by focusing on the endothelial compartment and its relative dysfunction in aging, which represents the main topic of the review. We modified the Introduction as follows: “The vascular endothelium, the active inner layer of the blood vessel, releases a wide array of biologically active molecules acting in an autocrine or paracrine fashion, thereby controlling arterial structure and vasodilatory, thrombolytic and vaso-protective func-tions. In particular, it regulates a number of biological functions, such as substrate ex-change/transport, innate immunity, the regulation of vascular tone by balancing the production of vasodilators and vasoconstrictors, angiogenesis, and hemostasis by se-creting antiplatelet and anticoagulant molecules (for a comprehensive review on the topic see (Xu et al, 2021). Endothelial dysfunction occurs during aging (Celermajer et al, 1994), where an imbalance between vasodilator and vasoconstriction factors, a progressive reduction of nitric oxide (NO) bioavailability, as well as an increase in cyclooxygenase (COX)-derived vasoconstrictor factors have been reported (Herrera et al, 2010; Celermajer et al, 1994). In addition, a decreased expression and activity of endothelial NOS (eNOS) has been observed in older animals (Smith et al, 2006). Moreover, endothelial dysfunction represents a risk factor for the development of several human vascular diseases, such as atherosclerosis, hypertension and stroke, peripheral arterial disease, metabolic syndrome (obesity, insulin resistance), and diabetes. The molecular mechanisms underpinning endothelial dysfunction are rather complex. Indeed, multiple mechanisms (i.e. impaired vasodilation, oxidative stress, inflammation, cell injury/death, senescence) have been reported to be involved. Notably, among them, decreased NO bioavailability represents a key hallmark of endothelial dysfunction. Thus, the severity of endothelial dysfunction has been shown to have prognostic value for cardiovascular events (Widlansky et al, 2003)”.

Unfortunately, the endothelial role of Pin1 in different clinical contexts, as mentioned by the Reviewer, is extremely limited and fully covered within the main text.

As requested by the Reviewer, we attach the graphical abstract summarizing the key message of the review (please see the attachment).

Reviewer 2 Report

  1. The title of the paper needs to be revised, the current one is a question. A more neutral title is required.
  2. Table 1, please provide in vitro and in vivo evidence of pin1 in other vascular cells, such as smooth muscle cells, macrophages. The evidence on endothelial cells is incomplete.
  3. introduction should starts with CVD and aging background.
  4. A figure of pin1 and pin family members domain structure is needed.
  5. Specify the site of eNOS phosphporylation in the abstract.
  6. Based on the evidence presented, the role of pin1 in vascular homeostasis and aging is unclear, more solid evidence can be presented and reviewed.
  7. A  recent review of endothelial function can be discussed and discuss whether Pin1 regulates other aspects of endothelial function. Pharmacol Rev. 2021 Jul;73(3):924-967.

Author Response

We thank the Reviewer for the comments and suggestions, that allowed us to greatly improve the paper.

Point 1: The title of the paper needs to be revised, the current one is a question. A more neutral title is required.

ANSWER: According the suggestions of the Reviewer, we modified the title of our review. The new title has been modified as follows: “Pin1 as molecular switch in vascular endothelium: notes on its putative in age-associated vascular diseases".

Point 2: Table 1, please provide in vitro and in vivo evidence of pin1 in other vascular cells, such as smooth muscle cells, macrophages. The evidence on endothelial cells is incomplete.

ANSWER: As suggested by the Reviewer we added in Table 1 data on Pin1 molecular targets in other vascular cells.

Point 3: Introduction should start with CVD and aging background.

ANSWER: As suggested by the Reviewer, we rewritten the introduction by focusing on the endothelial compartment and its relative dysfunction in aging, which represents the main topic of the review.

We modified the Introduction as follows: “The vascular endothelium, the active inner layer of the blood vessel, releases a wide array of biologically active molecules acting in an autocrine or paracrine fashion, thereby controlling arterial structure and vasodilatory, thrombolytic and vaso-protective func-tions. In particular, it regulates a number of biological functions, such as substrate ex-change/transport, innate immunity, the regulation of vascular tone by balancing the production of vasodilators and vasoconstrictors, angiogenesis, and hemostasis by se-creting antiplatelet and anticoagulant molecules (for a comprehensive review on the topic see (Xu et al, 2021). Endothelial dysfunction occurs during aging (Celermajer et al, 1994), where an imbalance between vasodilator and vasoconstriction factors, a progressive reduction of nitric oxide (NO) bioavailability, as well as an increase in cyclooxygenase (COX)-derived vasoconstrictor factors have been reported (Herrera et al, 2010; Celermajer et al, 1994). In addition, a decreased expression and activity of endothelial NOS (eNOS) has been observed in older animals (Smith et al, 2006). Moreover, endothelial dysfunction represents a risk factor for the development of several human vascular diseases, such as atherosclerosis, hypertension and stroke, peripheral arterial disease, metabolic syndrome (obesity, insulin resistance), and diabetes. The molecular mechanisms underpinning endothelial dysfunction are rather complex. Indeed, multiple mechanisms (i.e. impaired vasodilation, oxidative stress, inflammation, cell injury/death, senescence) have been reported to be involved. Notably, among them, decreased NO bioavailability represents a key hallmark of endothelial dysfunction. Thus, the severity of endothelial dysfunction has been shown to have prognostic value for cardiovascular events (Widlansky et al, 2003)”.

Point 4: A figure of pin1 and pin family members domain structure is needed.

ANSWER: We added in Figure 1 a ribbon representation of the full structure of human Pin1 (PDBID 1PIN; DOI: 10.1016/s0092-8674(00)80273-1), highlighting the PPIase and the WW domain in different colors (Figure 1A).

Point 5: Specify the site of eNOS phosphporylation in the abstract.

ANSWER: Following the suggestion of the Reviewer, we specified the site of eNOS phosphorylation in the abstract.

Point 6: Based on the evidence presented, the role of pin1 in vascular homeostasis and aging is unclear, more solid evidence can be presented and reviewed.

ANSWER: We agree with the observation of the Reviewer. Unfortunately, the endothelial role of Pin1 in different clinical contexts, as mentioned by the Reviewer, is extremely limited and fully covered within the main text.

Point 7: A recent review of endothelial function can be discussed and discuss whether Pin1 regulates other aspects of endothelial function. Pharmacol Rev. 2021 Jul;73(3):924-967.

ANSWER: We added in the introduction the reference suggested by the Reviewer, as follows: “In particular, it regulates a number of biological functions, such as substrate ex-change/transport, innate immunity, the regulation of vascular tone by balancing the production of vasodilators and vasoconstrictors, angiogenesis, and hemostasis by se-creting antiplatelet and anticoagulant molecules (for a comprehensive review on the topic see (Xu et al, 2021)” (page 1, lines 40-44). 

Round 2

Reviewer 2 Report

Previous concerns are well addressed and the manuscript is now acceptable.